# Taming Discrete Integration via the Boon of Dimensionality

**Jeffrey M. Dudek**
Rice University
Houston, USA

**Dror Fried**
The Open University of Israel
Israel

**Kuldeep S. Meel**
National University of Singapore
Singapore

## Abstract

Discrete integration is a fundamental problem in computer science that concerns the computation of discrete sums over exponentially large sets. Despite intense interest from researchers for over three decades, the design of scalable techniques for computing estimates with rigorous guarantees for discrete integration remains the holy grail. The key contribution of this work addresses this scalability challenge via an efficient reduction of discrete integration to model counting. The proposed reduction is achieved via a significant increase in the dimensionality that, contrary to conventional wisdom, leads to solving an instance of the relatively simpler problem of model counting. Building on the promising approach proposed by Chakraborty et al [9], our work overcomes the key weakness of their approach: a restriction to dyadic weights. We augment our proposed reduction, called DeWeight, with a state of the art efficient approximate model counter and perform detailed empirical analysis over benchmarks arising from neural network verification domains, an emerging application area of critical importance. DeWeight, to the best of our knowledge, is the first technique to compute estimates with provable guarantees for this class of benchmarks.

## 1 Introduction

Given a large set of items $S$ and a weight function $W$ that assigns weight to each of the items, the problem of *discrete integration*, also known as weighted counting, is to compute the weight of $S$, defined as the weighted sum of items of $S$. Often the set $S$ is implicitly represented, for example as the set of assignments that satisfy a given set of constraints. Such a representation allows one to represent an exponentially large set with polynomially many constraints, and thus capture other representation forms such as probabilistic graphical models [12, 40]. In particular, it is commonplace to implicitly represent $S$ as the set of assignments that satisfy a Boolean formula $F$ by using standard transformations from discrete domains to Boolean variables [6]. This allows one to leverage powerful combinatorial solvers within the process of answering the discrete integration query [24, 25, 8].

Discrete integration is a fundamental problem in computer science with a wide variety of applications covering partition function computations [23, 1], probabilistic inference [33, 51], network reliability estimation [20], and the verification of neural networks [5, 36, 37], which is the topic of this work. In particular, recent work demonstrated that determining properties of neural networks, such as adversarial robustness, fairness, and susceptibility to trojan attacks, can be reduced to discrete integration. Since discrete integration is #P-complete even when the weight function is constant [47], this has motivated theoreticians and practitioners alike to focus on approximation techniques.

Of particular interest in many settings, including neural network verification, is the case where the weight function can be expressed as *log-linear* distribution. Log-linear models are employed to capture a wide variety of distributions that arise from graphical models, conditional random fields, skip-gram models, and the like [28]. In the context of discrete integration, log-linear models are also

known as *literal-weighted*. Formally, the weight function $W$ assigns a weight to each variable $x_i$ such that $W(x_i) + W(\neg x_i) = 1$ (a variable or its negation is called a literal) [12]. Furthermore, the weight of an assignment $\sigma \in$ is defined as the product of all literals that are true in that assignment. For completeness, we formally state the equivalence between the two notions in Section 2. A special case of discrete integration is *model counting*, wherein the weight functions assigns a constant weight of $\frac{1}{2}$ to each literal. In this special case, the computation of the weight of $S$ reduces to computing the size of $S$, i.e. the number of solutions to the constraints. While from complexity theoretic viewpoint both discrete integration and model counting are #P-hard, tools for model counting have had significantly better scalability than those for discrete integration in practice [44, 17].

In this paper, we make a concrete progress towards answering discrete integration queries. For that, we follow the work of Chakbraborty et al. [9] that demonstrated a reduction from discrete integration to model counting. The general idea is to construct another formula $G$ from a discrete integration query $(F, W)$ such that that one can recover the weight of $F$ from the model count of $G$. We can then use a hashing-based tool [11, 44, 43] to compute an approximate model count of $G$. Finally, we use this count to compute an approximation to the weight of $F$ (with theoretical guarantees). Even though adding new variables in $G$ increases the dimensionality of the problem, we turn this seeming curse into a boon by benefiting from scalability of model counting techniques.

Although promising, the reduction of [9] has a key weakness: it can only handle weight functions where each literal weight is *dyadic*, i.e. the weight of each $x_i$ is of the form $\frac{k_i}{2^{m_i}}$ for some integers $k_i$ and $m_i$. Thus a pre-processing step is required to adjust each weight to a nearby dyadic weight. Moreover, the reduction of [9] simulates the weight function $W$ by adding variables and constraints in number proportional to $m_i$. Scaling to large queries such as those obtained in neural network verification thus entails using crude dyadic weights that poorly approximate the original query.

The primary technical contribution of this work is to lift the dyadic weights requirement. To this end, we introduce a new technique that can handle all rational weights, i.e. all weights of the form $W(x_i) = \frac{p_i}{q_i}$, while adding only an overhead of $\lceil \log_2(\max(p_i, q_i - p_i)) \rceil$ additional variables for each $x_i$. Thus, the pre-processing step of [9] is no longer required and and the designer has the ability to use accurate weights in the reduction that entail a complete guarantee for his estimates. A key strength of our approach is in the simplicity and the negligible runtime requirements of the reduction subroutine. When the overhead for every $x_i$ is, however, budgeted to $m_i$ additional variables, we present a new algorithm to compute the closest fraction $p/q$ to $W(x_i)$ such that $m_i \leq \log_2(p, q - p)$. Our algorithm employs an elegant connection to Farey Sequences [41].

Our technique results in a prototype implementation of our tool, called DeWeight. DeWeight employs the state-of-the-art approximate model counter ApproxMC [10, 11, 44] and allows theoretical $(\epsilon, \delta)$-guarantees for its estimates with no required weight adjustments. We perform a detailed evaluation on benchmarks arising from the domain of neural network verification.[1] Based on our empirical evaluation, DeWeight is, to the best of our knowledge, the first technique that can handle these benchmark instances while providing theoretical guarantees of its estimates. Our empirical results show that approximation of weights to the closest dyadic fractions leads to an astronomically large multiplicative error of over $10^8$ in even very simple weight settings, while our reduction is exact.

As a strong evidence to the necessity of our tool, recent work on neural network verification [5, 36, 37] was restricted to the uniformly weighted case (i.e., where for all $x_i$, $W(x_i) = \frac{1}{2}$) due to unavailability of scalable techniques for discrete integration; the inability of existing (weighted) discrete integration tools to handle such benchmarks is well noted, and noted as promising future work in [5]. Our work thus contributes to broadening the scope of neural network verification efforts.

**Related work.**   Recently, hashing-based techniques have emerged as a dominant paradigm in the context of model counting [46, 26, 10, 25, 50, 11, 2, 44, 43]. The key idea is to reduce the problem of counting the solutions of a formula $F$ to that of polynomially many satisfiability (SAT) queries wherein each query is described as the formula $F$ conjuncted with random XOR constraints. The concurrent development of CryptoMiniSat [45, 44, 43], a state of the art SAT solver with native support for XORs, has led to impressive scalability of hashing-based model counting techniques.

In a significant conceptual breakthrough, Ermon et al. [25] showed how hashing-based techniques can be lifted to discrete integration. Their framework, WISH, reduces the computation of discrete

integration to linearly many optimization queries (specifically, MaxSAT queries) wherein each query is invoked over $F$ conjuncted with random XOR constraints. The unavailability of MaxSAT solvers with native support for XORs has hindered the scalability of WISH style techniques. A dual approach proposed by Chakraborty et al. [8, 16] reduces discrete integration to linearly many counting queries over formulas wherein each of counting query is constructed by conjunction of $F$ with Pseudo-Boolean constraints, which is shown to scale poorly [39]. In summary, there exists a wide gap in the theory and practice of hashing-based approaches for discrete integration.

In this context, instead of employ hashing-based approaches for discrete integration, we rely on our reduction to model counting, albeit via introduction of additional variables, thereby increasing the dimensionality of the problem. One would normally expect that such an increase in the dimensionality would not be a wise strategy, but our empirical evaluation bears evidence of the boon of dimensionality. Therefore, our strategy allows augmenting the reduction with state of the art approximate model counter such as ApproxMC [10, 11, 44]. While we employ ApproxMC in our empirical analysis, we note that our reduction can be combined with any other state of the art approximate model counter.

**Organization** The rest of the paper is organized as follows: We provide notations and preliminaries in Section 2. We then introduce in Section 3 the primary contribution of this paper: an efficient reduction from discrete integration to model counting that allows for non-dyadic weights. Since the designer can still choose to budget the reduction extra variables by weight adjustment, we utilize in Section 4 our approach to construct a new algorithm that employs a connection to Farey sequences [41] to compute the closest fraction to $W(x_i)$ with a limited number of bits. This is followed by a detailed empirical evaluation in Section 5. Finally, we conclude in Section 6. Detailed proofs for the lemmas and theorems in the paper appear in the appendix of the full version of the paper, see [18].

## 2   Notations and Preliminaries

We are given a Boolean formula $F$ with a set of variables $X = \{x_1, x_2, \ldots x_n\}$ that appear in $F$. A *literal* is a variable or it's negation. A *satisfying assignment*, also called *witness* of $F$, is an assignment of the variables in $X$ to either *true* or *false*. We denote by $\sigma(x)$ the value assigned to the variable $x$ by the assignment $\sigma$, and denote by $\sigma^1 = \{i \mid \sigma(x_i) = true\}$ and $\sigma^0 = \{i \mid \sigma(x_i) = false\}$ the set of variables assigned *true* and respectively *false* by $\sigma$. We denote the set of all witnesses of $F$ by $R_F$.

As mentioned earlier, we focus on the log-linear discrete integration problem in which weights are assigned to literals and the weight of an assignment is the product of weights of its literals. For a variable $x_i$ of $F$ and a weight function $W$, we use $W(x_i)$ and $W(\neg x_i)$ to denote the non-negative, real-valued weights of the positive and negative literals. As is standard, we assume without loss of generality that $W(x_i) + W(\neg x_i) = 1$; weights where $W(x_i) + W(\neg x_i) \neq 1$ can be easily changed to sum 1 with normalization. To ease the notations, we overload $W$ to denote the weight of a literal, assignment or formula, depending on the context. Thus, for an assignment $\sigma$ we denote $W(\sigma) = \prod_{i \in \sigma^1} W(x_i) \prod_{i \in \sigma^0} W(\neg x_i)$. The reader may observe that literal-weighted formulation is an equivalent representation of log-linear models [28]. In particular, by viewing $\sigma(X) \in \{0, 1\}^n$, we have $\Pr[X] \propto e^{\theta \cdot X}$ for $\theta \in \mathbb{R}^n$ such that $\theta[i] = \log\left(\frac{W(x_i)}{W(\neg x_i)}\right)$.

Given a set $Y$ of assignments, we denote $\sum_{\sigma \in Y} W(\sigma)$ by $W(Y)$. For a formula $F$, we use $W(F)$ to denote $\sum_{\sigma \in R_F} W(\sigma)$. Since a weight $W(x_i) = 1$ (respectively $W(x_i) = 0$) can always be reduced to non-weight by adding the unit clause $(x_i)$ (respectively the unit clause $(\neg x_i)$) to $F$, we assume w.l.o.g. that $0 < W(x_i) < 1$ for every variable $x_i$.

An instance of the log-linear discrete integration problem can be defined by a pair $(F, W)$, where $F$ is a Boolean formula, and $W$ is the weight function for literals, and returns $W(F)$. The discrete integration problem is also referred to as weighted model counting [8]. We say that a probabilistic algorithm $\mathcal{A}$ computes $(\varepsilon, \delta)$ estimate of $W(F)$, if for a *tolerance* $\varepsilon > 0$ and a *confidence* $1 - \delta \in [0, 1)$, the value $v$ returned by $\mathcal{A}$ satisfies $\Pr[\frac{W(F)}{1+\varepsilon} \leq v \leq (1+\varepsilon) \cdot W(F)] \geq 1-\delta$. A crucial ingredient of the approached proposed in this paper is the problem of model counting. Given a formula $F$, the problem of approximate model counting returns $v$ such that $\Pr[\frac{R_F}{1+\varepsilon} \leq v \leq (1 + \varepsilon) \cdot R_F] \geq 1 - \delta$.

We are often interested in assignments over a subset of variables $P \subseteq X$. $P$ is called *projection set* or *sampling set*. We use $R_{F \downarrow P}$ to indicate the projection of $R_F$ on $P$, which is the set of all assignments $\sigma'$ to $P$ for which there exists some $\sigma \in R_F$ that agrees with $\sigma'$ on all of $P$. Given a weight function $W$ defined over the variables of $P$, we abuse notation and define $W(F \downarrow P) = \sum_{\sigma' \in R_{F \downarrow P}} W(\sigma')$. As has been noted in recent work [21], projection is a theoretically powerful formulation and recent applications of counting and integration have advocated usage of projection [5, 36, 37]. For the sake of clarity, we present our techniques without considering projection but the technical framework developed in this paper easily extends to the projected formulation; see the appendix in [18] for details.

## 3   From Discrete Integration to Model Counting

In this section, we focus on the reduction of discrete integration to model counting. Our reduction is built on top of the approach of Chakraborty et al [9]. Their work, however, is restricted to the setting where the weight of each $x_i$ is dyadic, i.e., of the form $W(x_i) = \frac{k_i}{2^{m_i}}$ where $k_i$ is a positive odd number less than $2^{m_i}$. We begin by revisiting the notion of *chain formulas* introduced in [9].

Let $m > 0$ and $k < 2^m$ be natural numbers[2] . Let $c_1 c_2 \cdots c_m$ be the $m$-bit binary representation of $k$, where $c_m$ is the least significant bit. We then construct a chain formula $\varphi_{k,m}(\cdot)$ with exactly $k$ satisfying assignments over $m$ variables $a_1, \ldots a_m$ as follows. Let $t(k)$ be the last bit that is 1, i.e., $c_{t(k)} = 1$ and $c_j = 0$ for all $j > t(k)$. For every $j$ in $\{1, \ldots m-1\}$, let $C_j$ be the connector "$\vee$" if $c_j = 1$, and the connector "$\wedge$" if $c_j = 0$. Then we define

$$\varphi_{k,m}(a_1, \cdots a_m) = a_1 \, C_1 \, (a_2 \, C_2 (\cdots (a_{t(k)-1} \, C_{t(k)-1} \, a_{t(k)}) \cdots))$$

We call $\varphi_{k,m}$ a *chain formula* due to its syntactic nature of a variable chain. Consider for example $k = 10, m = 4$. Then we have the binary representation of $k$ to be 1010 and $\varphi_{10,4}(a_1, a_2, a_3, a_4) = (a_1 \vee (a_2 \wedge a_3))$. Note that while $\varphi_{10,4}$ is defined over $\{a_1, a_2, a_3, a_4\}$, $a_4$ does not appear in $\varphi_{10,4}$.

Now, let $\varphi_{k_i, m_i}$ be a chain formula with variables $(x_{i,1}, x_{i,2}, \ldots x_{i,m_i})$. The core conceptual insight of Chakraborty et.al. [9] was to observe that if all the variables$\{x_{i,1}, x_{i,2}, \ldots x_{i,m_i}\}$ are assigned a weight of $\frac{1}{2}$ then the weight of $\varphi_{k_i, m_i}$ is $\frac{k_i}{2^{m_i}}$. As noted in Section 2, when every variable is assigned weight of $\frac{1}{2}$, the problem of discrete integration reduces to model counting. Therefore, one can reduce discrete integration to model counting by replacing every occurrence of $x_i$ by $\varphi_{k_i, m_i}$. This is achieved by adding the clauses $(x_i \leftrightarrow \varphi_{k_i, m_i})$, where the variables of $\varphi_{k_i, m_i}$ are *fresh* variables (that is, do not appear anywhere else in the formula), and assigning weight of $\frac{1}{2}$ to every variable.

Formally, [9] showed a transformation of a discrete integration query $(F, W)$ to a model counting problem of a formula $\widehat{F}$ such that $W(F) = C_W \cdot |R_{\widehat{F}}|$ wherein $C_W = \prod_{x_i \in X} 2^{-m_i}$. The above reduction crucially utilizes the property that the weight is represented as dyadic number $\frac{k_i}{2^{m_i}}$ and thus forces a pre-processing step where weights such as $1/3$ or $2/5$ are adjusted to dyadic representations. Since scalability requires a limited number of added variables, adjustments can be highly inaccurate, leading to orders of magnitude difference in the final estimates. In this work, we avoid these adjustments by proposing an efficient transformation to handle general weights of the form $p/q$.

**Lifting the dyadic number restriction.**   We next show how to handle the cases when $W(x_i) = \frac{p_i}{q_i}$ where $p_i < q_i$ are arbitrary natural numbers. Rather than replace $x_i$ with its corresponding chain formula, the key insight is a usage of implications that allows us to simulate both the weight of $x_i$ and $\neg x_i$ *separately* by using two different chain formulas.

Consider again the reduction of [9]. The formula $\varphi_{k_i, m_i}$ used for a weight $W(x_i) = k_i/2^{m_i}$, has $m_i$ fresh variables and a solution space of size $2^{m_i}$. The formula $\Omega_i = (x_i \leftrightarrow \varphi_{k_i, m_i})$ has $k_i$ solutions, out of $2^{m_i}$ total, when $x_i$ is *true*, simulating the weight $k_i/2^{m_i}$. Similarly $\Omega_i$ has $2^{m_i} - k_i$ solutions, when $x_i$ is *false*, simulating the weight $1 - k_i/2^{m_i}$.

The observation that we make is that since, for each variable $x$, there is no assignment that can satisfy both $x$ and $\neg x$, we have the freedom to use two separate chain formulas associated with $x$

and $\neg x$. Thus, for every non-negative integers $k_i, k_i', m$ where $k_i, k_i' \leq 2^m$, the formula $\Omega_i = (x_i \rightarrow \varphi_{k_i,m_i}) \wedge (\neg x_i \rightarrow \varphi_{k_i',m_i})$ has $k_i$ solutions when $x_i$ is *true* and $k_i'$ solutions when $x_i$ is *false*. That is, $k_i/(k_i + k_i')$ fraction of the solutions of $\Omega_i$ have $x$ assigned to *true* and $k_i'/(k_i + k_i')$ fraction of the solutions of $\Omega_i$ have $x$ assigned to *false*. Note that $\varphi_{k_i,m_i}$ and $\varphi_{k_i',m_i}$ can even use the same sets of variables since there are no assignments in which both $x_i$ and $\neg x_i$ are *true*. The only limitation is that $2^m \geq \max(k_i, k_i')$, i.e. $m_i \geq \lceil \max(\log_2 k_i, \log_2 k_i') \rceil$.

For example, consider the formula with a single variable $F(x_1) = x_1$ and a weight function with $W(x_1) = p/q$ for $p < q$. Set $m = \lceil \max(\log_2 p, \log_2(q - p)) \rceil$ (that is, the smallest $m$ for which both $p < 2^m$ and $q - p < 2^m$). Observe that the model count of $F \wedge \Omega$, where $\Omega = (x_1 \rightarrow \varphi_{p,m}) \wedge (\neg x_1 \rightarrow \varphi_{q-p,m})$, can be used to compute $(F, W)$. This is because there are $p$ solutions of $\Omega$ in which $x_1$ is assigned *true*, $q - p$ solutions of $\Omega$ in which $x_1$ is assigned *false*, and $p + q - p = q$ solutions of $\Omega$ total. Restating, $p/q$ fraction of the solutions of $\Omega$ have $x_1$ assigned to *true*, corresponding to the weight $W(x_1) = p/q$. Similarly, $(q - p)/q$ fraction of the weights of $\Omega$ have $x_1$ assigned to false, corresponding to the weight $W(\neg x_1) = (q - p)/q$. Then, since $W(F) = \frac{p}{q}$, we have that the model count of $F \wedge \Omega$ is $p = 1 \cdot \frac{p}{q} = q \cdot W(F)$.

It is worth noting that since the same variables can be used to describe the chain formulas for $p$ and $q - p$, the number of bits to describe $p$ and $q - p$ can be less than the number of bits to describe the denominator $q$. For example for $W(x_i) = \frac{1}{3}$, we have that $m_i = \lceil max(\log_1 1, \log_2 2) \rceil = 1$ even though the denominator is greater than 2.

We formalize all this in the following lemma.

**Lemma 1.** *Let* $(F, W)$ *be a given discrete integration instance such that* $W(x_i) = \frac{p_i}{q_i}$ *and* $W(x_i) + W(\neg x_i) = 1$ *for every* $i$. *Let* $m_i = \lceil \max(\log_2 p_i, \log_2(q_i - p_i)) \rceil$, *and let* $\hat{F} = F \wedge \Omega$, *where* $\Omega = \bigwedge_i ((x_i \rightarrow \varphi_{p_i,m_i}) \wedge (\neg x_i \rightarrow \varphi_{q_i-p_i,m_i}))$. *Denote* $C_W = \prod_{x_i} q_i$. *Then* $W(F) = \frac{|R_{\hat{F}}|}{C_W}$.

**Discrete integration estimate algorithm.** We observe that our reduction is also approximation preserving. This is specifically because we lifted the restriction to dyadic weights and thus removed the extra source of error in [9] that comes from adjusting weights to be dyadic. Thus, our overall approximate algorithm $\mathcal{A}$ for discrete integration works as follows. Given a discrete integration query $(F, W)$, together with a *tolerance* $\varepsilon > 0$ and a *confidence* $1 - \delta \in [0, 1)$, $\mathcal{A}$ first constructs the formula $\hat{F}$ and normalization $C_W$ with Lemma 1. It then feeds $(\hat{F}, \varepsilon, \delta)$ to an approximated model counter $\mathcal{B}$, divides the result by $C_W$ and returns that value. We then have the following theorem.

**Theorem 1.** *The return value of* $\mathcal{A}(F, \varepsilon, \delta)$ *is an* $(\varepsilon, \delta)$ *estimate of* $W(F)$. *Furthermore,* $\mathcal{A}$ *makes* $\mathcal{O}\left( \frac{\log(n + \sum_i m_i) \log(1/\delta)}{\varepsilon^2} \right)$ *calls to an* NP *oracle, where* $m_i = \lceil \max(\log_2 p_i, \log_2(q_i - p_i)) \rceil$.

*Proof.* The proof is deferred to the full version [18]. ∎

## 4 Pre-Processing of Weights via Farey Sequences

The experimental results shown below in Section 5 demonstrate that many benchmarks are still challenging for approximate discrete inference tools. We observe that the scalability of the model counting techniques depends on the number of variables. In this context, when faced with a large benchmark, we may need to approximate weights to minimize the number of additional variables introduced for each variable $x_i$. As we observed in Section 5, a naive approximation strategy such as approximating to the closest dyadic weights may introduce unacceptably large error. We seek to design a strategy to minimize the error introduced due to such an approximation.

For $m > 0$ and integers $a < b$ we call the fraction $a/b$ an $m$-*bit fraction* if $a/b$ is irreducible and both $a \leq 2^m$ and $b - a \leq 2^m$. A nearest $m$-bits fraction is then an $m$-bits fraction $p/q$ such that $|\frac{a}{b} - \frac{p}{q}| \geq |\frac{\hat{p}}{\hat{q}} - \frac{p}{q}|$ for every $m$-bit fraction $\frac{a}{b}$. Given a pair of integers $(p, q)$, where $p < q$, we seek to find a nearest $m$-bit fraction to $\frac{p}{q}$.

For example if $(p, q) = (4, 25)$ (hence $q - p = 21$) then we have that $1/6$ is the nearest 3-bit fraction to $4/25$, and significantly closer to $p/q$ than the corresponding nearest 3-bit dyadic fraction $1/8$.

**Algorithm 1** An algorithm for finding a nearest $m$-bit fractions to $p/q$.

---

**Input**: Fractions $(a_1, b_1), (p, q), (a_2, b_2)$, non-negative integer $m$
**Output**: Fractions $(c_1, d_1)$

1:  $a \leftarrow a_1 + a_2$
2:  $b \leftarrow b_1 + b_2$
3:  **if** $(a_1, b_1) = (p, q)$ or $(a_2, b_2) = (p, q)$ or $(a, b) = (p, q)$ **then**
4:     **return** $(p, q)$
5: **else if** $\lceil \log_2(a) \rceil > m$ or $\lceil \log_2(b - a) \rceil > m$ **then**
6:     **return** $\min\{(a_1, b_1), (a_2, b_2)\}$
7: **else if** $a/b < p/q$ **then**
8:     **return** ApproxFraction$((a, b), (p, q), (a_2, b_2))$
9: **else if** $p/q < a/b$ **then**
10:     **return** ApproxFraction$((a_1, b_1), (p, q), (a, b))$
11: **end if**

---

We present Algorithm 1 that finds a nearest $m$-bit fraction as Algorithm 1, where $(a, b)$ represents a fraction $a/b$. Algorithm 1 exploits features of the Farey sequence [41], where the Farey sequence $\mathcal{F}_k$ is the sequence (in increasing order) of all irreducible fractions with denominators of at most $k$. The key fact is that for every two consecutive fractions $a_1/b_1, a_2/b_2$ in $\mathcal{F}_k$ we have that $(a_1 + a_2)/(b_1 + b_2)$ is in $\mathcal{F}_{k+1}$. Since all the $m$-bits fraction are members of $\mathcal{F}_{2^m - 2}$, properties of the Farey sequence [41] guarantee us that at every step nearer $m$-bits fractions to $p/q$ (from bottom and top) are the next input to Algorithm 1.

**Theorem 2.** *Algorithm 1 with initial arguments* $(p, q)$, $(a_1, b_1) = (0, 1)$ *and* $(a_2, b_2) = (1, 1)$ *finds a nearest* $m$-*bit fraction to* $p/q$.

*Proof.* The proof is deferred to the full version [18]. $\qquad\qquad\qquad\qquad\qquad\qquad\qquad\qquad\qquad\square$

The maximal running time of Algorithm 1 is $O(2^{2m} - 2)$. The following example shows that this can also be a worst case. Consider any input $1/q$ where $q > 2^m$ and $m$ is the number of the required bits. In such case at every step we have that $a_1/b_1 = 0/1$, and so $a/b = a_2/(b_2 + 1)$. These examples thus requires $2^{2m} - 2$ steps.

## 5   Implementation and Evaluation

We implemented our reduction framework, without the pre-processing step of Section 4, in a tool called DeWeight in 800 LOC of C++ and a 100 LOC Python wrapper. DeWeight receives as input a weighted discrete integration query $(F, W)$ and performs the reduction in Section 3 to produce an unweighted (projected) model-counting instance $\hat{F}$. Following our Algorithm $\mathcal{A}$ in Section 3, DeWeight then invokes the approximate model counter ApproxMC4 [43] to obtain a $(\epsilon, \delta)$ estimate to the model count of $\hat{F}$. DeWeight computes from this an $(\epsilon, \delta)$ estimate to $W(F)$. In keeping in line with the prior studies, we configure a tolerable error $\epsilon = 0.8$ and confidence parameter $\delta = 0.2$ as defaults throughout the evaluation.

**Benchmarks.** We consider a set of 1056 formulas from [5] arising from the domain of neural network verification. Each formula is a projected Boolean formula in conjunctive normal form (CNF) on the order of $10^5$ variables total, of which no more than 100 are in the sampling set. Variables in the sampling set correspond to bits of the input to a binarized neural network, while variables not in the sampling set correspond to intermediate values computed during the network inference. Each formula is satisfied if a specific property holds for that input in the neural network.

There are three classes of formulas considered in [5], quantifying *robustness* (960 formulas), *trojan attack effectiveness* (60 formulas), and *fairness* (36 formulas). We outline each briefly here but refer the reader to [5] for a complete description. Formulas for robustness and trojan attack effectiveness are based on a binarized neural network trained on the MNIST [31] dataset. Each of these formulas has a sampling set of 100 variables interpreted as pixels in a $10 \times 10$ binarized input image. Formulas for robustness are satisfied if the input image encodes an adversarial perturbation from a fixed test

set image. Formulas for trojan attack effectiveness are satisfied if the input image has a fixed trojan trigger and the output is a fixed target label. Formulas for fairness are based on a binarized neural network trained on the UCI Adult dataset [3]. Each of these formulas has a sampling set of 66 variables interpreted as 66 binary features. Formulas for fairness are satisfied if the classification of the network changes when a sensitive feature is changed.

Due to lack of efficient techniques to handle discrete integration, Baluta et al. [5] and Narodytska [37] settled for empirical studies restricted to uniform distribution, i.e., these formulas were unweighted and their model count estimates the probability that the property holds when inputs are drawn from a uniform distribution over all inputs. As stated earlier, our formulation allows us to model log-linear distributions. To this end, we assign weights to variables in the sampling set, and consequently the resulting discrete integration queries compute the probability that the property holds when the input is drawn from the specified log-linear distribution.

**Comparison approaches.** We implemented the dyadic-only reduction of [9] in C++ and refer to it as Dyadic. Given a weighted discrete integration query $(F, W)$ and a number of bits $k$, Dyadic adjusts the weight of each literal to the closest dyadic weight that can be expressed in $k$ bits or less. This produces a new weighted discrete integration query $(F, W')$ where all weights are dyadic. The result $W'(F)$ of this new query can be seen as a $(\gamma_{W,k}, \delta = 0)$ estimate to $W(F)$, where $\gamma_{W,k} \in [0, 1]$ is an extra error term that can be computed explicitly from $W$ and $k$. Dyadic then reduces $(F, W')$ to an unweighted projected-model-counting instance and invokes ApproxMC4. Overall, Dyadic produces an $(\epsilon \cdot \gamma_{W,k} + \gamma_{W,k} + \epsilon, \delta)$ estimate to $W(F)$. This estimate is no better than the estimate produced by DeWeight but often much worse.

Although there are also a variety of guarantee-less techniques for discrete integration [38, 22, 48, 49, 34, 32], we focus here on techniques that provide theoretical guarantees on the quality of results. As noted above, the benchmarks we consider from neural network verification are both large (with on the order of $10^5$ variables) and projected. This makes them challenging for many techniques. Many exact tools, including c2d [13] and D4 [29], do not efficiently support projection and so our neural network verification benchmarks are observed to be beyond their capabilities. [3] The state-of-the-art technique is DSharp [35], owing to its ability to perform projection with D2C [4]. DSharp can compile a projected discrete integration query into d-DNNF [14]. This step is independent of the weights, and takes most of the total running time. A d-DNNF reasoner [7] can then answer the projected discrete integration query. This produces the exact answer to the query, not just an $(\epsilon, \delta)$ estimate. We also consider GANAK [42], which computes a $(0, \delta)$ estimate (i.e., computes the exact answer to the query with probability $1 - \delta$). Finally, we implemented the WISH algorithm [25] using MaxHS [15, 27] as a black box MaxSAT solver; this results in a $(15, \delta)$ estimate to the discrete integration theory.

We also considered evaluating these benchmarks with WeightMC [8], but found WeightMC unsuitable for these benchmarks for the following reason. WeightMC can only handle benchmarks with a small *tilt*, which is the ratio of the largest weight assignment to the smallest weight assignment. This is because WeightMC requires an upper bound $r$ on the tilt to be provided as input; the running time of WeightMC is then linear in $r$ (see Theorem 2 of [8]). In particular, the runtime is lower bounded by the minimum of tilt and $2^n$, where $n$ is the size of sampling set. The bound on tilt, however, is extremely large for the benchmarks that we consider. For example, in Experiment 1 a benchmark with a sampling set of $n$ variables has tilt (a priori) upper bounded by $\frac{(2/3)^n}{(1/3)^n} = 2^n$; the bound on tilt for our benchmarks is never smaller than $2^{66}$.

**Experimental Setup.** All experiments are performed on 2.5 GHz CPUs with 24 cores and 96GB RAM. Each discrete integration query ran on one core with a 4GB memory cap and 8 hour timeout.

## 5.1 Evaluation

Of the unweighted 1056 formulas, 126 are detected as unsatisfiable by ApproxMC within 8 hours and so we discard these. We consider two different weight settings on the remaining satisfiable benchmarks. Times recorded for DeWeight and Dyadic include both the time for the reduction stage and time spent in ApproxMC; the time for the reduction stage was never more than 4 seconds.

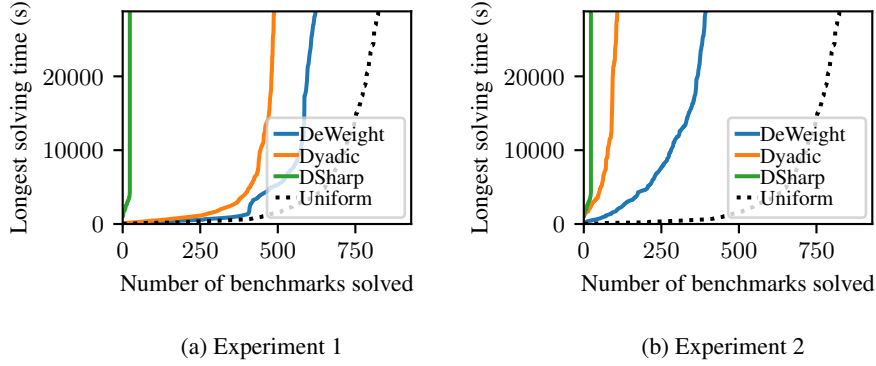

(a) Experiment 1  (b) Experiment 2

Figure 1: A comparison of the running times of DeWeight and Dyadic against the compilation time of DSharp and the time for the uniformly weighted version of each query. Each $(x, y)$ point on a line indicates that the tool was able to answer $x$ queries using no more than $y$ seconds per query.

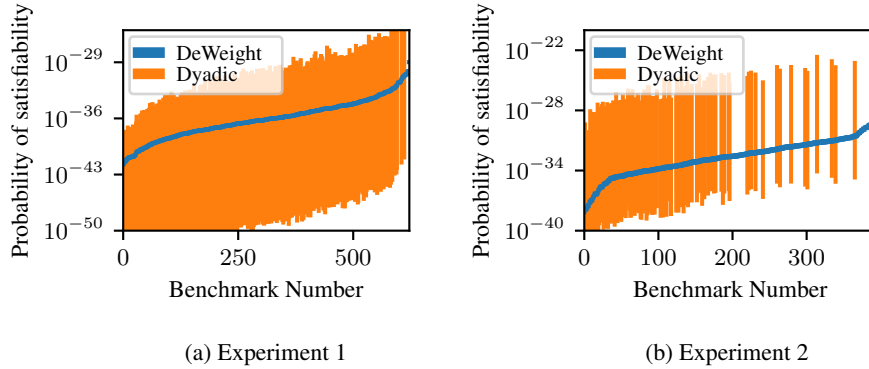

(a) Experiment 1  (b) Experiment 2

Figure 2: The guarantees returned by DeWeight and Dyadic. Each vertical line on each plot represents the interval $[A, B] \subseteq \mathbb{R}$ returned by a tool on a given discrete integration query. The intervals are ordered on the x-axis by the center of the interval returned by DeWeight, and drawn for Dyadic only when it completes within 8 hours. Each tool guarantees with confidence $\delta = 0.2$ that the true answer to the query is contained within the corresponding interval. We observe that the intervals for DeWeight are always much smaller than the intervals for Dyadic, indicating that DeWeight returns much tighter guarantees.

**Experiment 1: Identical weights on all variables.** We first consider setting weights $W(x) = 2/3$ and $W(\neg x) = 1/3$ on every variable $x$. These weights represent the "best-case scenario" for DeWeight, since DeWeight is able to represent such weights using a single extra bit per variable. We ran Dyadic using 2 extra bits per variable. In this case Dyadic adjusts each weight to $W(x) = 3/4$ and $W(\neg x) = 1/4$. We also ran Dyadic with 3 extra bits per variable, but it timed out on all benchmarks.

Results on the computation time of these queries are presented in Figure 1a. Within 8 hours, DeWeight solved 622 queries while Dyadic solved 489. There was only 1 query that Dyadic was able to solve while DeWeight could not. Both GANAK and WISH were not able to solve any queries. We conclude that DeWeight is significantly faster than the competing approaches in this setting.

We also present a comparison of the quality of the estimates produced by DeWeight and Dyadic in Figure 2a. We observe the estimates produced by DeWeight are significantly more accurate than those produced by Dyadic. On each of these benchmarks, the weight adjustment required by Dyadic results an additional error term $\gamma$ for Dyadic that is always at least $1.76 \cdot 10^8$. Consequently, while DeWeight always returns a $(0.8, 0.2)$ estimate with our settings, on each of these queries Dyadic never returned better than a $(3.17 \cdot 10^8, 0.2)$ estimate.

**Experiment 2: 1 decimal weights on all variables.** We next consider setting weights where each $W(x)$ is chosen uniformly and independently at random from $\{0.1, 0.2, \cdots, 0.9\}$ (and $W(x) + W(\neg x) = 1$). DeWeight requires $22/9 \approx 2.44$ extra bits per variable in expectation for this setting. We ran Dyadic with at most 3 bits per variables; this results in adjusting down to the nearest factor of $1/8$ (e.g., $4/10$ is adjusted to $3/8$) and using $19/9 \approx 2.11$ extra bits per variable in expectation.

Results on the computation time of these queries are presented in Figure 1b. Both GANAK and WISH were not able to solve any queries. We find that DeWeight is able to solve 394 queries in this setting while Dyadic is only able to solve 108 (and solved no queries that DeWeight did not also solve). This is somewhat surprising, since Dyadic uses fewer bits per variable in expectation than DeWeight. We hypothesize that this difference is because the chain formulas in DeWeight are easier for ApproxMC than those in Dyadic. It is worth remarking that over 99% of the time inside ApproxMC is spent in the underlying SAT calls. Understanding the behavior of SAT solvers, particularly on SAT formulas with XOR clauses, is considered a major open problem in SAT community [19].

We also present a comparison of the estimates produced by DeWeight and Dyadic in Figure 2b. Even when Dyadic is able to solve the benchmark we observe that DeWeight is significantly more accurate, since 1 decimal weights are difficult to approximate well using 3 bit dyadic weights.

## 6 Conclusion

Discrete integration is a fundamental problem in computer science with a wide variety of applications. In this work, we build on the paradigm of reducing discrete integration to model counting and propose a reduction that relaxes the restriction to dyadic weights from prior work. A salient strength of our work is that it enables us to build on the recent progress in the context of model counting [46, 26, 10, 25, 50, 11, 2, 44, 43]. The resulting tool, DeWeight, is able to handle instances arising from neural network verification domains, a critical area of concern. These instances were shown to be beyond the ability of prior work.

One limitation of our framework is that due to the performance of the underlying approximate model counter, our method is currently limited to work well only for problems where the problem can be projected on 100 or fewer variables (note however that our method can handle many – on the order $10^5$ – variables that are not in the sampling set). Even with this limitation, however, DeWeight is the only technique that is able to handle such benchmarks with large tilt. Furthermore, we trust that improvements in the underlying approximate model counter, which is itself a highly active research area, will allow for larger sampling sets in the future.

Another limitation is, as we showed in Theorem 1, the performance of our method depends heavily on the total number of bits required for the weight description. The number of bits must currently be relatively small (e.g., up to 3 decimal places) for our method to perform well. Nevertheless, our experiments indicate that the prior state of the art approaches are unable to handle weights with even one decimal place on large benchmarks.

Finally, as noted before, our current focus in this paper is on benchmarks arising from the verification of binarized neural networks. Although we do not consider other domains, our benchmark suite comprises of over 1000 formulas from three different applications: robustness, trojan attack effectiveness, and fairness. Furthermore, since the underlying approximate counter, ApproxMC, has been used on a wide variety of benchmarks and given DeWeight's runtime is dictated by ApproxMC's runtime performance, we expect DeWeight to be useful in other contexts as well.

## Acknowledgements

We are grateful to Supratik Chakraborty and Moshe Vardi for many insightful discussions. This work was supported in part by National Research Foundation Singapore under its NRF Fellowship Programme [NRF-NRFFAI1-2019-0004] and AI Singapore Programme [AISG-RP-2018-005], by NUS ODPRT Grant [R-252-000-685-13], and by NSF grants [IIS-1527668, CCF-1704883, IIS-1830549, and DMS-1547433]. The computational work for this article was performed on resources of the National Supercomputing Centre, Singapore `https://www.nscc.sg`. Any opinions, findings and conclusions or recommendations expressed in this material are those of the author(s) and do not reflect the views of National Research Foundation, Singapore.

## Broader Impact

Discrete integration is a fundamental problem in machine learning and therefore, it is critical to develop algorithmic techniques that provide rigorous formal guarantees and can handle real-world instances. Our work in this paper takes a significant step, in our view, towards achieving this goal. Our work does not make use of bias in the data.

## Footnotes

[1]All code is available in a public repository at `https://github.com/meelgroup/deweight`

[2]A careful reader may observe that the above definition is a minor adaptation of the original definition [9] and no longer requires $k$ to be odd.

[3]We also considered ProjMC [30] but the Linux kernel on our systems is too old to run the published precompiled tool. We were unable to get either the code or a different precompiled version from the authors.

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
