[Supplementary Material]

# A  Full Proofs

## A.1   Proof for Lemma 1

**Lemma.** *Let $(F, W(\cdot))$ be a given discrete integration instance such that $W(x_i) = \frac{p_i}{q_i}$ and $W(x_i) + W(\neg x_i) = 1$ for every $i$. Let $m_i = \lceil \max(\log_2 p_i, \log_2(q_i - p_i)) \rceil$, and let $\hat{F} = F \wedge \Omega$, where $\Omega = \bigwedge((x_i \rightarrow \varphi_{p_i, m_i}) \wedge (\neg x_i \rightarrow \varphi_{q_i - p_i, m_i}))$. Denote $C_W = \prod_{x_i} q_i$. Then $W(F) = \frac{|R_{\hat{F}}|}{C_W}$.*

*Proof.* Note that if $W(x_i) = p_i/q_i$ then $W(\neg x_i) = (q_i - p_i)/q_i$. Let $W'(\cdot)$ be a new weight function, defined over the literals of $X$ as follows. $W'(x_i) = p_i$ and $W'(\neg x_i) = q_i - p_i$. Note that $W'(\cdot)$ is different from the typical weight functions considered in this paper as $W'(x_i)$ and $W'(\neg x_i)$ are non negative integers. By extending the definition of $W'(\cdot)$ in a natural way (as was done for $W(\cdot)$) to assignments, sets of assignments and formulas, it is easy to see that $W(F) = W'(F)/C_W$.

Next, for every assignment $\sigma$ of variables in $X$, we have that $W'(\sigma) = \prod_{i \in \sigma^1} p_i \prod_{i \in \sigma^0}(q_i - p_i)$. Let $\hat{\sigma}$ be an assignment of variables appearing in $\hat{F}$. We say that $\hat{\sigma}$ is *compatible* with $\sigma$ if for all variables $x_i$ in the original set of variables $X$, we have $\hat{\sigma}(x_i) = \sigma(x_i)$. Observe that $\hat{\sigma}$ is compatible with exactly one assignment of variables in $X$. For every assignment $\sigma$ for $F$, let $S_\sigma$ denote the set of all satisfying assignments of $\hat{F}$ that are compatible with $\sigma$. Then $\{S_\sigma | \sigma \in R_F\}$ is a partition of $R_{\hat{F}}$. From the chain-formula properties, we know that there are $p_i$ witnesses of $\varphi_{p_i, m_i}$ and $q_i - p_i$ witnesses of $\varphi_{q_i - p_i, m_i}$. Since the representative formulas of every weighted variable use a fresh set of variables, and since there is no assignment that can make both a variable and it's negation to become true, we have from the structure of $\hat{F}$ that if $\sigma$ is a witness of $F$, then $|S_\sigma| = \prod_{i \in \sigma^1} p_i \prod_{i \in \sigma^0}(q_i - p_i)$. Therefore $|S_\sigma| = W'(\sigma)$. Note that if $\sigma$ is not a witness of $F$, then there are no compatible satisfying assignments of $\hat{F}$; hence $S_\sigma = \emptyset$ in this case. Overall, this gives

$$|R_{\hat{F}}| = \sum_{\sigma \in R_F} |S_\sigma| + \sum_{\sigma \notin R_F} |S_\sigma| = \sum_{\sigma \in R_F} |S_\sigma| + 0 = W'(F).$$

It follows that $W(F) = \frac{W'(F)}{C_W} = \frac{|R_{\hat{F}}|}{C_W}$. $\qquad\square$

We note that the number $m_i$ is picked, only so the truth table of $\varphi_{p_i, m_i}$ can store $p_i$ assignments, and that the truth table of $\varphi_{q_i - p_i, m_i}$ can store $q_i - p_i$ assignments.

## A.2   Proof for Theorem 1

**Theorem.** *The return value of $\mathcal{A}(F, \varepsilon, \delta)$ is an $(\varepsilon, \delta)$ estimate of $W(F)$. Furthermore, $\mathcal{A}$ makes $\mathcal{O}\left(\frac{\log(n + \sum_i m_i) \log(1/\delta)}{\varepsilon^2}\right)$ calls to an NP oracle, where $m_i = \lceil \max(\log_2 p_i, \log_2(q_i - p_i)) \rceil$.*

*Proof.* Denote the return value that the approximated model counter $\mathcal{B}$ returns by $v$. Then we have that $\Pr[\frac{|R_{\hat{F}}|}{1+\varepsilon} \leq v \leq (1 + \varepsilon)|R_{\hat{F}}|] \geq 1 - \delta$. By dividing the returned value $v$ by the factor $C_W$ we then have that $\Pr[\frac{|R_{\hat{F}}|}{C_W(1+\varepsilon)} \leq \frac{v}{C_W} \leq (1 + \varepsilon)\frac{R_{\hat{F}}}{C_W}] \geq 1 - \delta$. Recall that from Lemma 1 we have that $W(F) = \frac{|R_{\hat{F}}|}{C_W}$. Then since $\mathcal{A}(F, \varepsilon, \delta)$ returns $v' = v/C_W$, we all in all have that $\Pr[\frac{W(F)}{1+\varepsilon} \leq v' \leq (1 + \varepsilon)W(F)] \geq 1 - \delta$. That is, the return value $\mathcal{A}(F, \varepsilon, \delta)$ is an $(\varepsilon, \delta)$ estimate of $W(F)$ as required.

The number of NP oracle calls made by Algorithm $\mathcal{A}$ follows from Theorem 4 of [11], and the fact that $\hat{F}$ has $n + \sum_i m_i$ variables ($n$ from the original formula and $\sum_i m_i$ added in the chain formulas). $\qquad\square$

## A.3   Handling projected formulation

For the sake of clarity, we presented our techniques without considering projection. However, since the underlying model counter that we use, ApproxMC [10, 11, 43], handles projected model counting, the technical framework described in this paper can be easily extended to the projected formulation.

To see that, note that the formulation of a projected weighted Boolean formula $F$ is $(F, P, W)$ where $P$ is a projected set, and the weight function $W$ is defined only over the variables of $P$. Our algorithm $\mathcal{A}$ reduces $(F, P, W)$ using the chain formula reduction of Lemma 1, to a projected unweighted Boolean formula $(\hat{F}, P \cup Y)$, where $Y$ denotes the set of fresh variables used for the chain formulas. $(\hat{F}, P \cup Y)$ is then fed to ApproxMC that supports projected model counting. The result value $v$ that ApproxMC returns is an $(\varepsilon, \delta)$ estimate to $(\hat{F}, P \cup Y)$. It follows that $\mathcal{A}$ returns $v/C_W$ as an $(\varepsilon, \delta)$ estimate to $(F, P, W)$.

## A.4 Proof for Theorem 2

As shorthand, in this section we use $bin(a)$ to denote the binary representation of $a$ and $|bin(a)|$ to denote the number of bits that are needed to describe $a$ (i.e., $\lceil \log_2(a) \rceil$).

**Theorem.** *Algorithm 1 with initial arguments $(p, q)$, $(a_1, b_1) = (0, 1)$ and $(a_2, b_2) = (1, 1)$ finds a nearest $m$-bit fraction to $p/q$.*

*Proof.* First note that since the required $p$ and $q - p$ are of size $m$ bits at most, the denominator of a potential nearest $m$-bits fraction is no bigger than $k = 2^m - 2$. Therefore, the required $p/q$ is contained in the Farey Sequence $\mathcal{F}_k$, that is the sequence of all irreducible fractions (in increasing order) with denominator of at most size $k$.

A way to construct the entire Farey sequence $\mathcal{F}_i$ from $\mathcal{F}_{i-1}$ is as follows: Initially set $\mathcal{F}_i = \mathcal{F}_{i-1}$. Then iteratively go over the members of $\mathcal{F}_i$ in an increasing order, and for every $a_1/b_1 < a_2/b_2$ neighbors in $\mathcal{F}_{i-1}$, construct $(a_1+a_2)/(b_1+b_2)$. It turns out that $(a_1+a_2)/(b_1+b_2)$ is an irreducible fraction and that $a_1/b_1 < (a_1+a_2)/(b_1+b_2) < a_2/b_2$. Now, if $b_1+b_2 = i$, add $(a_1+a_2)/(b_1+b_2)$ to $\mathcal{F}_i$, otherwise skip. Finally arrange $\mathcal{F}_i$ in an increasing order. The initial sequence is $\mathcal{F}_1 = (0/1, 1/1)$. Then for example $\mathcal{F}_2 = (0/1, 1/2, 1/1)$, $\mathcal{F}_3 = (0/1, 1/3, 1/2, 2/3, 1/1)$ and so on.

The algorithm ApproxFraction follows the Farey sequence construction by setting at every call $a = a_1 + a_2$, $b = b_1 + b_2$, and evaluating $a/b$. Assume that both $|bin(a)|$ and $|bin(b - a)|$ are at most than $m$. Then $a/b$ is a candidate for the nearest $m$-bit fraction to $p/q$, where Lines 7-10 check whether $|a/b - p/q| < |a_1/b_1 - p/q|$ or $|a/b - p/q| < |a_2/b_2 - p/q|$, and makes the recursive call replacing either $a_1/b_1$ with either $a/b$ if $a/b$ is $m$-bit nearer from the bottom or either replacing $a_2/b_2$ with $a/b$ if $a/b$ is $m$-bit nearer from the top.

Algorithm ApproxFraction bounds to stop as the denominator always increases (i.e. $b_1 + b_2 > b_1$ and $b_1 + b_2 > b_2$). It is left to see that when the algorithm stops, the value of $\min\{(a_1, b_1), (a_2, b_2)\}$ is the $m$-nearest fraction to $p/q$. First, if either $a_1/b_1$, $a_2/b_2$ or $a/b$ is equal to $p/q$, then the algorithm returns $p/q$ in Line 4 which is obviously the $m$-bits nearest fraction. Next, assume that either $|bin(a)|$ or $|bin(b - a)|$ are bigger than $m$ as the stopping condition in Line 5 indicates. consider the interval $(a_1/b_1, a_2/b_2)$. Since the nearest $m$-bits fractions are members of $\mathcal{F}_k$, these must be found via the Farey sequence construction above. Since for every $i$, only consecutive fractions of $\mathcal{F}_i$ are used to generate members of $\mathcal{F}_{i-1}$, it follows that the the nearest $n$-bits fraction must be generated, as in the Farey sequence construction, by using only fractions from the interval $(a_1/b_1, a_2/b_2)$. We show by induction that for every $i$, there are no $m$-bit fractions in $\mathcal{F}_i \cap (a_1/b_1, a_2/b_2)$. First set $\mathcal{F}_j$ to be the Farey sequence for which both $a_1/b_1, a_2/b_2 \in \mathcal{F}_j \backslash \mathcal{F}_{j-1}$ Then $a_1/b_1, a_2/b_2$ are consecutive in $\mathcal{F}_j$, therefore $\mathcal{F}_j \cap (a_1/b_1, a_2/b_2)$ is empty. Assume by induction that for every $i \geq i$, $\mathcal{F}_i \cap (a_1/b_1, a_2/b_2)$ does not contain $m$-bits fraction. Now, observe that $\mathcal{F}_{i+1} \cap (a_1/b_1, a_2/b_2)$ is generated from consecutive fractions in $\mathcal{F}_i \cap [a_1/b_1, a_2/b_2]$. This can be done only if the two fractions are $a_1/b_1, a_2/b_2$, and then we had that the fraction $a/b = (a_1+a_2)/(b_1+b_2)$ is not an $m$-bit fraction, or otherwise at least one of the fractions belongs to $\mathcal{F}_i \cap (a_1/b_1, a_2/b_2)$, hence is not an $m$-bits fraction. The following lemma shows that in this case as well, the result is not an $m$-bit fraction, hence all in all $\mathcal{F}_{i+1} \cap (a_1/b_1, a_2/b_2)$ is empty as well. As such, the nearest $m$-bits fractions from bottom and top are $(a_1/b_1)$ and $(a_2/b_2)$ respectively and algorithm returns $\min\{(a_1/b_1), (a_2/b_2)\}$, which is the nearest $m$-bits fraction as required. $\square$

**Lemma 2.** *Let $a/b$, $x/y$ be a fraction where $0 < a < b$, $0 < x < y$ and either $|bin(a)|$ or $|bin(b-a)|$ are bigger than $n$. Consider the fraction $(a+x)/(b+y)$. Then either $|bin(a+x)| > m$ or $|bin((b+y) - (a+x))| > m$.*

*Proof.* Obviously for every two numbers $i, j$, $i < j \iff bin(i) < bin(j) \iff |bin(i)| < |(bin(j)|$. Assume $|bin(a)| > m$. Then since $x$ is positive then $(a + x) > a$. Thus $|bin(a + x)| > m$. Next, assume $|bin(b - a)| > m$. Then since $y - x > 0$ then $(b + y) - (a + x) = (b - a) + (y - x) > (b - a)$, so $|bin((b + y) - (a + x))| > m$. $\qquad\square$

Finally from the analysis above of the stopping conditions of $ApproxFraction$, we have that the maximal running time of $ApproxFraction$ is $2^{2m} - 2$. The following example shows that this can also be a worst case. Consider any input $1/q$ where $q > 2^m$ and $m$ is the number of the required bits. In such case at every step we have that $a_1/b_1 = 0/1$, and so $a/b = a_2/b_2 + 1$. This gives an overall running time of $2^{2m} - 2$.