[Reviews · NeurIPS 2020]

Review 1

Summary and Contributions: In this paper, the authors proposed a new method for reducing discrete integration problems to simpler model counting problems, that relaxes the restriction of dyadic weights. The proposed method is evaluated in the context of benchmarks from neural network verification, and the effectiveness of the method is validated.

Strengths: The idea of reducing discrete integration to model counting is not new. However, the contribution of relaxing the dyadic weight restriction is significant, and the need for pre-processing steps (which could also result in poor approximations) is lifted. The theoretical aspects of this work seem solid and provide rigorous guarantees of the proposed method.

Weaknesses: Unfortunately, I cannot comment on the scientific contribution as well as the weakness of the paper, as I do not possess the expertise to judge the theoretical results accurately. My expertise is so outside of this field that I will rely on the judgement of the other reviewers, whom I hope will have more experience and will better know the literature.

Correctness: I cannot judge the correctness of theoretical results since I am not an expert in this field.

Clarity: Yes.

Relation to Prior Work: Yes.

Reproducibility: Yes

Additional Feedback:


Review 2

Summary and Contributions: This paper builds upon the WMC to MC reduction presented in [1], proposing a technique that lifts the restriction of the original work to dyadic weights. Leveraging an existing (epsilon,delta)-approximate model counter ApproxMC4 [2], the resulting (epsilon,delta)-approximate WMC solver, dubbed DeWeight, is tested on benchmarks arising from neural network verification [3].

Strengths: Although the proposed technique is evaluated on formal verification tasks only, the problem is relevant for a much larger number of applications. The proposed technique effectively addresses the limitations of a previously proposed approach.

Weaknesses: I think that the paper could be more explicit in describing the limits of the proposed approach. My main criticism regards the empirical evaluation, which is In my opinion, it could be improved (see detailed comments).

Correctness: The approach looks correct, although I am not entirely convinced by the empirical evaluation of the solver (see detailed comments).

Clarity: Yes.

Relation to Prior Work: Yes.

Reproducibility: Yes

Additional Feedback: I think that the proposed reduction technique is a neat improvement over [1], although I find it difficult to assess the significance of this contribution. Given the generality of DeWeight, I wonder why the authors evaluated it only on benchmarks from [4]. I am aware that WISH-like approaches suffer the lack of XOR support in MaxSAT solvers and that some (epsilon,delta)-approximate WMC solvers do not natively support projected MC, but the choice of benchmarks and competitors is rather restricted and makes it hard to answer the ultimate question "How good is DeWeight at WMC?". Is there a reason for not including [4] in the comparison? After reading the paper I felt that a number of questions were left unanswered: is DeWeight a state-of-the-art solver for WMC or should I use it only when my problems can be projected on few Boolean variables? How well does the proposed reduction scale with respect to existing WMC approaches when the weights are arbitrary rationals and not in {0.1, 0.2, ..., 0.9}? Since very often the priors are learned from data, why not sampling the weights uniformly from [0,1]? Finally, "In particular, recent work demonstrated that determining properties of neural networks, such as adversarial robustness, fairness, and susceptibility to trojan attacks, can be reduced to discrete integration." I think that it applies to NNs with discrete inputs only, such as the binarized NNs from [4]. I wish the paper was more explicit in describing the limitations of the approach. --- [1] Chakraborty, Supratik, et al. "From weighted to unweighted model counting." Twenty-Fourth International Joint Conference on Artificial Intelligence. 2015. [2] Soos, Mate, Stephan Gocht, and Kuldeep S. Meel. "Accelerating approximate techniques for counting and sampling models through refined CNF-XOR solving." Proceedings of International Conference on Computer-Aided Verification (CAV). Vol. 7. 2020. [3] Baluta, Teodora, et al. "Quantitative verification of neural networks and its security applications." Proceedings of the 2019 ACM SIGSAC Conference on Computer and Communications Security. 2019. [4] Chakraborty, Supratik, et al. "Distribution-aware sampling and weighted model counting for SAT." Twenty-Eighth AAAI Conference on Artificial Intelligence. 2014. --- I thank the authors for their feedback. The size of the tilt clarifies why [4] was not considered on that particular experiment. I too agree that the small tilt assumption is not realistic and I wish that this aspect was mentioned in the text. DeWeight makes assuptions on the weight function too. The results on the NN verification benchmarks are impressive but the setting feels carefully chosen to exclude a priori any other competitor (other than Dyadic). I agree that 1000+ formulas represent a fairly large benchmark, but they come from the same application domain (thus having similar structural properties) and the choice of weights is rather limited. I still think that the empirical evaluation could be extended to other settings (even synthetic ones) and include other solvers. Some natural questions to ask: how does DeWeight relate to existing WMC approaches when the weights are arbitrary? When the reduction to (unweighted) MC can be leveraged in practice? The discussion of the proposed approach tends to overlook these limitations. After reading the rebuttal I slightly raised my score. Provided that the authors clarify the limitations of the proposed technique, this submission could be accepted with the current experimental section. Yet, I hope that the authors will provide more empirical evidence of the merits and limitations of the proposed reduction.


Review 3

Summary and Contributions: 1. it builds on the paradigm of reducing discrete integration to model counting and propose a reduction that relaxes the restriction to dyadic weights from prior work. 2. it addresses this scalability challenge via an efficient reduction of discrete integration to model counting.

Strengths: It proposes a reduction that relaxes the restriction to dyadic weights from prior work with theoratical guarantees and with some empirical evaluation.

Weaknesses: 1. This work claims its much better scalability than previous work. But the expriments itself is not large-scale. 2. This work claims that it's useful for neural network verification domains but I don't see experiments on these domains. 3. Baseline methods are a little few and experiments here are not enough to support its argument.

Correctness: I am not quite sure.

Clarity: I think there is still much room for the authors to refine their paper.

Relation to Prior Work: Yes.

Reproducibility: Yes

Additional Feedback:

[Author Response · NeurIPS 2020]

We are deeply appreciative of the reviewers for their feedback amidst these trying circumstances. We are glad that the reviewers appreciate the scalability challenge addressed by our work, and the neatness of our proposed algorithm. We would like to emphasize that we compared with over 1056 benchmarks arising from the domain of neural network verification.

## Reviewer 3

**Comparison with WeightMC**  We indeed considered evaluating these benchmarks with WeightMC [1], as the reviewer suggests, and found WeightMC unsuitable for our benchmarks for the following reason. WeightMC can only handle benchmarks with a small *tilt*, which is the ratio of the largest weight assignment to the smallest weight assignment. In particular, WeightMC requires an upper bound $r$ on the tilt to be provided as input; the running time of WeightMC is then linear in $r$ (see Theorem 2 of [1]). In particular, the runtime is lower bounded by the minimum of tilt and $2^n$, where $n$ is the size of sampling set. The bound on tilt, however, is extremely large for the benchmarks that we consider. For example, in Experiment 1 a benchmark with a sampling set of $n$ variables has tilt (a priori) upper bounded by $\frac{(2/3)^n}{(1/3)^n} = 2^n$; the bound on tilt for our benchmarks is never smaller than $2^{66}$. Therefore, WeightMC can not handle any of the benchmarks. To summarize, DeWeight is indeed the state of the art technique for benchmarks with large tilt. To the best of our knowledge, we are not aware of any practical applications of discrete integration that have small tilt.

**Projection on few variables**  Due to the performance of the underlying approximate model counter, our methods is currently limited to work well only for problems where the problem can be projected on 100 or fewer variables (note however that our method can handle many – on the order $10^5$ – variables that are not in the sampling set). Even with this limitation, DeWeight is the only technique that is able to handle such benchmarks with large tilt. Furthermore, we trust that improvements in the underlying approximate model counter, which is itself a highly active research area, will allow for larger sampling sets in the future.

**Other Weight Settings**  As we showed in Theorem 1, the performance of our method depends heavily on the total number of bits required for the weight description. Thus one of the limitations our approach is that this total number of bits should be relatively small (say, for up to 3 decimal places). As our experiments indicate, the prior state of the art approaches are unable to handle weights with even one decimal place.

**Limitations**  We will make it clear that we focus on benchmarks arising from the verification of binarized neural networks. Please note that our benchmark suite comprises of over 1000 formulas, a fairly large dataset comprising of three different applications: robustness, trojan attack effectiveness, and fairness. Furthermore, since the underlying approximate counter, ApproxMC, has been shown to be used in wide variety of benchmarks and given DeWeight's runtime is dictated by ApproxMC's runtime performance, we expect DeWeight to be useful in other contexts as well.

## Reviewer 4

As mentioned on line 219, we tested our tool on 1056 formulas arising from the domain of neural network verification. These formulas evaluate robustness, trojan attack effectiveness, and fairness of a binarized neural network. Our experiments showed how on these benchmarks we outperform state-of-the-art tools. Please note that we have compared the tool with the state of the art techniques (as mentioned, WISH and WeightMC can not handle these instances, and therefore, we did not add plots corresponding to WISH and WeightMC).

## References

[1] S. Chakraborty, D. J. Fremont, K. S. Meel, S. A. Seshia, and M. Y. Vardi. Distribution-aware sampling and weighted model counting for SAT. In *Proc. of AAAI*, pages 1722–1730, 2014.


[Meta-Review · NeurIPS 2020]

This is a borderline paper that offers material that is of interest to the community. That said, the experimental results could be more robust and this is the main reason for the borderline assessment.